# Chiral Separation of Oxazolidinone Analogs by Capillary Electrophoresis Using Anionic Cyclodextrins as Chiral Selectors: Emphasis on Enantiomer Migration Order

**DOI:** 10.3390/molecules28114530

**Published:** 2023-06-02

**Authors:** Zoltán-István Szabó, Francisc Boda, Béla Fiser, Máté Dobó, Levente Szőcs, Gergő Tóth

**Affiliations:** 1Faculty of Pharmacy, George Emil Palade University of Medicine, Pharmacy, Science, and Technology of Targu Mures, Gh. Marinescu 38, 540139 Târgu Mureș, Romania; zoltan.szabo@umfst.ro (Z.-I.S.); francisc.boda@umfst.ro (F.B.); 2Sz-imfidum Ltd., Lunga nr. 504, 525401 Covasna, Romania; 3Higher Education and Industrial Cooperation Centre, University of Miskolc, Egyetemváros, H-3515 Miskolc, Hungary; bela.fiser@uni-miskolc.hu; 4Ferenc Rákóczi II. Transcarpathian Hungarian Institute, 90200 Beregszász, Transcarpathia, Ukraine; 5Department of Physical Chemistry, Faculty of Chemistry, University of Lodz, 90-149 Łódź, Poland; 6Department of Pharmaceutical Chemistry, Semmelweis University, Hőgyes E. 9, H-1085 Budapest, Hungary; dobomate99@gmail.com; 7Cyclolab Ltd., Illatos út 7, H-1097 Budapest, Hungary; szocs@cyclolab.hu

**Keywords:** oxazolidinone, reversal of enantiomeric migration order, cyclodextrin, CE, enantioseparation, chiral building block

## Abstract

Comparative chiral separations of enantiomeric pairs of four oxazolidinone and two related thio-derivatives were performed by capillary electrophoresis, using cyclodextrins (CDs) as chiral selectors. Since the selected analytes are neutral, the enantiodiscrimination capabilities of nine anionic CD derivatives were determined, in 50 mM phosphate buffer pH = 6. Unanimously, the most successful chiral selector was the single isomeric *heptakis*-(6-sulfo)-β-cyclodextrin (HS-β-CD), which resulted in the highest enantioresolution values out of the CDs applied for five of the six enantiomeric pairs. The enantiomer migration order (EMO) was the same for two enantiomeric pairs, irrespective of the CD applied. However, several examples of EMO reversals were obtained in the other cases. Interestingly, changing from randomly substituted, multi-component mixtures of sulfated-β-CD to the single isomeric chiral selector, enantiomer migration order reversal occurred for two enantiomeric pairs and similar observations were made when comparing *heptakis*-(2,3-di-O-methyl-6-O-sulfo)-β-CD, (HDMS-β-CD) with HS-β-CD. In several cases, cavity-size-dependent, and substituent-dependent EMO reversals were also observed. Minute differences in the structure of the analytes were also responsible for several cases of EMO reversal. The present study offers a complex overview of the chiral separation of structurally related oxazolidinones, and thio-analogs, highlighting the importance of the adequate choice of chiral selector in this group of compounds, where enantiomeric purity is of utmost importance.

## 1. Introduction

Given that there are usually pharmacological and/or pharmacokinetic differences between individual enantiomers, the importance of stereochemistry is gaining greater attention in drug research and development [1]. There are primarily two main approaches to obtain enantiomerically pure compounds: chiral resolution and asymmetric synthesis, by using enantiomerically pure substrates, reagents, solvents, or catalysts [2]. Among the various protocols developed to induce chirality to a substrate, the use of chiral auxiliaries stands out as offering high levels of stereoselectivity and reliability [3]. Their role in asymmetric reactions is comparable to those of different protection and deprotection reactions in organic synthesis [3]. Thus, chiral auxiliaries are employed temporarily to convey chirality to a prochiral substrate via simple reactions to obtain the desired stereochemistry of transformations [2,4].

Chiral oxazolidine-2-ones are probably among the most commonly used auxiliaries in asymmetric syntheses [3,5]. First disclosed by Evans et al. in 1981 [6], oxazolidinones have found numerous applications, and there are several reviews that discuss their vital role in the asymmetric synthesis of structurally diverse compounds [3,4,5].

Apart from their widespread use as chiral auxiliary agents, the oxazolidine-2-one nucleus also plays a central role in the structure of several biologically active molecules, including antimicrobials, monoamine oxidase inhibitors, protein or enzyme inhibitors, and occasionally, they can also be found in the framework of natural products [7,8]. Oxazolidinones as a class of antibiotics act by inhibiting protein synthesis by binding to the ribosomal RNA of the 50S subunit without affecting the 30S subunit [5]. Linezolid, the most well-known representative of this class, was the first oxazolidinone approved for clinical use by the Food and Drug Administration (FDA) in 2000. A second representative, tedizolid, was approved in 2014 by the FDA, while contezolid was approved in 2021 by the National Medical Products Administration of China [9], and several other members of this class (such as sutezolid [10], ranezolid [11], delpazolid [12] etc.) are under clinical development.

In both presented application areas (chiral auxiliaries and antibiotics), the optical purity of the oxazolidinones is crucial. Chirality plays a significant role in oxazolidinone antibiotics, as many of these drugs demonstrate stereoselective effects. For example, linezolid contains a chiral center at the C-3 position of its oxazolidinone ring, and the *S*-enantiomer is approximately four times more potent than the *R*-enantiomer in inhibiting bacterial growth.

Given the importance of chirality in this class of compounds, development of effective enantioseparation methods is crucial to ensure the quality, safety, and efficacy of the obtained products. Enantioseparations are still dominated by liquid chromatography using chiral stationary phases. This approach has been accepted as the industry standard not only for the determination of enantiomeric purity of active drug substances and pharmaceutical products but also for (semi)-preparative scale isolation of individual enantiomers [13,14,15]. Capillary electrophoresis (CE) clearly has some limitations compared to chromatographic techniques, the most obvious being its lower sensitivity and reproducibility. However, due to some of its unique features, it is still an extremely desirable alternative technique for chiral separations. The obvious advantages of CE methods over chromatography include high separation efficiency, short analysis time, and low reagent and sample consumption. Apart from these, the underlying reason for the notable disparities between chromatographic and electrophoretic methods is the migration principle of the analyte in CE. The major dissimilarities between these techniques stem from the featured selectivity of electrophoretic migration for species within the same physical phase. Additionally, in direct liquid chromatographic enantioseparations on chiral stationary phases, the selector is immobilized, whereas, in CE, the selector–analyte complex is generally mobile [16,17].

Undoubtedly, cyclodextrins (CDs) are the preferred chiral selectors used in CE [18,19,20,21]. This is mainly attributed to their UV transparency, low toxicity, favorable aqueous solubility, cost-effectiveness, and the abundance of various derivatives with diverse physicochemical properties and enantiorecognition ability. Concerning their chemical structure, CDs are cyclic oligosaccharides, α-1,4-linked *D*-glucopyranose units. Their spatial structure resembles an open truncated cone, characterized by a hydrophilic outer shell and a lipophilic inner cavity. The size of the cavity varies depending on the number of *D*-glucopyranose units, with α-CD containing six units, β-CD containing seven units, and γ-CD containing eight units.

There are some examples of enantioseparations of analytes containing oxazolidinone motifs in their structure using CDs as chiral selectors in capillary electrophoresis. A method based on CD-mediated micellar electrokinetic chromatography was developed for the determination of the enantiomeric purity of PHA-549184, an oxazolidinone antibiotic. Among the five sulfated CDs tested, high-sulfated γ-CD (HS-γ-CD) was selected as the chiral selector. Due to the low aqueous solubility of the active, use of a non-ionic surfactant (Brij35) was also needed for proper method performance [22]. Comprehensive studies were performed by Michalska et al. on the chiral separation of several oxazolidinone antibiotics. The enantiomeric purity of linezolid was determined using *heptakis*-(2,3-diacetyl-6-sulfato)-β-CD (HDAS-β-CD) as chiral selector [23], and in a subsequent study, NMR and molecular modeling were employed to elucidate the chiral recognition process [24]. For the enantioseparation of another oxazolidinone antibiotic, tedizolid, charged single isomeric CDs were employed. The best results were obtained with the same HDAS-β-CD as chiral selector in 50 mM formic buffer pH 4.0, with the addition of acetonitrile [25]. The same research group also performed the enantiomeric separation of radezolid using four single isomeric sulfated β-CD derivatives. Baseline resolution of the enantiomers was observed when using *heptakis*-(2,3-di-O-methyl-6-O-sulfo)-β-CD (HDMS-β-CD) in phosphate buffer pH 2.5, with the addition of acetonitrile [26]. Nonaqueous CE, using either HDAS-β-CD or HDMS-β-CD in formate buffer-based acetonitrile–methanol containing background electrolyte (BGE) was successful for the enantioseparation of sutezolid. Complementary techniques, such as infrared and NMR spectroscopy, alongside molecular modeling, were employed to provide information about the transient diastereomer complex formation between the enantiomers and the chiral selector [27].

Although all the above-mentioned analytes contain an oxazolidine core in their structure, none of these studies deal with the enantioseparation of the smaller, substituted oxazolidinones and thio-derivatives, usually used as chiral auxiliaries.

Continuing and further expanding our studies regarding enantiomer elution/migration order reversals among oxazolidinone analogues [28], in the present work, the comparative chiral separation of the enantiomers of four oxazolidinones and two, structurally similar thio-analogs were performed using various anionic CDs as chiral selectors. Special attention was given to the effects of differences in the cavity size, side chain, and degree of substitution (DS) of different CDs on the enantiomer separation performance and especially to the enantiomer migration order. Differences arising in enantiomer recognition patterns due to the analogs’ structure were also considered.

## 2. Results and Discussion

### 2.1. General Overview of the Enantioseparations

The structure of the studied oxazolidinones, and related thio-analogs are depicted in Figure 1. All analytes are neutral under typical CE conditions (background electrolyte (BGE) with pH 2–10), with the predicted p*K*_a_ values (MolGpka [29]) ranging between 12.1 (compound **1**) and 13.1 for the oxazolidine-2-thione derivate (compound **6**).

In the case of neutral analytes, charged CDs have the advantage of possessing electrophoretic self-mobility, enabling them to separate uncharged compounds in addition to the chiral separation of charged compounds. Among the charged CDs, anionic derivatives are more frequently applied for chiral separations compared to their cationic counterparts. This is due to their cost-effectiveness in production, and wider availability from various vendors. They also offer electrostatic interaction and countercurrent mobility with the predominantly cationic pharmaceutical active ingredients. While certainly versatile, especially for the enantioseparation of acidic analytes, cationic derivatives can also alter the direction of the electro-osmotic flow (EOF) because they can dynamically coat the inner wall of the capillary, which can either be an advantage or can add further complexity to enantioseparations [30].

In the present study, the applicability of nine different anionic CDs were tested for the chiral separation of the selected analytes, namely: carboxymethyl-α-CD (CM-α-CD), carboxymethyl-β-CD (CM-β-CD), carboxymethyl-γ-CD (CM-γ-CD), carboxyethyl-β-CD (CE-β-CD), sulfobutylether-β-CD (SBE-β-CD), succinyl-β-CD (Succ-β-CD), sulfated-β-CD (S-β-CD); *heptakis*-(6-O-sulfo)-β-CD (HS-β-CD), and *heptakis*-(2,3-di-O-methyl-6-O-sulfo)-β-CD (HDMS-β-CD). Apart from the latter two, the anionic CDs employed in this study are randomly substituted, multi-component mixtures with a declared average DS (degree of substitution, the average number of derivatized hydroxyl groups). HS-β-CD and HDMS-β-CD are well-defined, single-isomer CD derivatives.

Initial separations were performed in 50 mM phosphate buffer pH 6.0, at 25 °C, using CD concentrations between 2.5–25 mM, at 5–6 concentration levels with a separation voltage of 20 kV. In cases where the generated current permitted it (<100 μA), 50 mM concentrations of CDs were also employed while reducing the separation voltage to 15 kV. A summary of the 54 different separation cases is presented in Table 1, including the maximal enantioresolution value (*R*_s_ max.), and the observed enantiomer migration order (EMO). Using the abovementioned conditions baseline separations (*R*_s_ > 1.5) were achieved in 17 cases, while enantiorecognition in form of at least peak splitting was observed in 42 cases.

HS-β-CD was the most efficient anionic CD for the selected analytes. This single-isomer sulfated CD provided baseline separation for five of the six studied enantiomeric pairs and provided high *R*_s_ values for the enantiomers of compounds **1**, **4**, **5**, and **6** (*R*_s_ > 3). This finding is noteworthy in comparison to the results reported by Michalska et al., where no, or only weak, enantiorecognition was observed for linezolid [26], radezolid [26], or tedizolid [25] when using HS-β-CD. However, it should be noted that the mentioned antibiotics are characterized by a bulkier structure than the studied oxazolidinones herein.

The least well-performing derivatives were the carboxyalkyl CDs (CM-α-CD, CM-β-CD, CM-γ-CD, and CE-β-CD). Apart from CM-β-CD, which separated the enantiomers of compounds **4**, **5**, and **6**, none of the others offered baseline separation for any of the studied compounds. CM-α-CD and CM-γ-CD displayed particularly low enantiodiscrimination capabilities towards the selected oxazolidinone analogs, as chiral interactions were only scarcely observed.

When comparing the results obtained for the different analytes, it is interesting to note that almost all CDs showed chiral interactions with the enantiomers of compounds **4**, **5**, **6**, while the enantiomers of compound **2** showed the least cases of enantiodiscrimination, and the obtained *R*_s_ values were also generally the lowest among the studied oxazolidinone derivatives.

### 2.2. Cavity-Size-Dependent EMO Reversal

By comparing the enantioseparation results achieved with various carboxymethylated CD derivatives possessing different cavity sizes, it was found that CM-β-CD yielded the most favorable results. This finding suggests that the specific cavity of β-CD derivatives is the most suitable size for separating the analyzed compounds. As mentioned earlier, while CM-β-CD, showed chiral interactions with all compounds, baseline separation was observed only for the enantiomers of compounds **5**, and **6**. However, it must be noted that with further optimization—which was not the goal of the present work—baseline separation may also be achievable for other compounds as well. Cavity-size-dependent EMO reversals were observed in several cases for the studied compounds when using the CM-α-, β-, and γ-CD, indicating that there are differences in chiral recognition for the different cavity-sized selectors towards the enantiomers of the studied analytes.

No separation occurred for the enantiomers of compound **2** when using CM-α-CD, even up to a chiral selector concentration of 50 mM. On the other hand, as shown in Figure 2A, when using CM-β-CD, the migration order was **2*S***, followed by **2*R***, while upon changing to the larger cavity-sized selector (CM-γ-CD), EMO reversal was observed, **2*R*** being the first migrating enantiomer. A similar trend was observed for the thio-analog, compound **4**, where EMO reversal was observed when changing the chiral selector from CM-β-CD to CM-γ-CD (Figure 2B).

In the case of compound **3**, no EMO reversal was observed when using carboxymethylated CDs with different cavity sizes. However, this was the only analog which displayed chiral interactions with CM-α-CD. EMO in the case of the narrower cavity-sized chiral selector was the same as for CM-β-CD, ***3R***, followed by ***3S***.

The scientific literature has reported differences in the chiral recognition of various CD derivatives with differing cavity sizes towards a diverse range of analytes [31,32,33,34,35,36,37,38]. In cases where NMR studies were also performed alongside CE enantioseparations, the opposite EMOs were rationalized based on the differences in the spatial structures of the transient diastereomeric complexes [34,35,36,37].

### 2.3. Substituent-Dependent EMO Reversal

A key benefit of using CDs as chiral selectors in CE is their potential for derivatization. This process involves the introduction of various noncharged and charged groups in a random or selective manner on the CD rims. This derivatization can not only enhance the chiral recognition ability of the native CD derivative but can also induce EMO reversal. Multiple instances of substituent-dependent EMO reversals were observed while investigating the chiral separation of the analytes, and these are discussed in detail.

In this section, a comparison is made between EMO acquired using different β-CD derivatives that vary in their substituents. It is worth noting that HS-β-CD is not included in this analysis as the EMO reversals seen between S-β-CD and HS-β-CD are discussed in a subsequent section.

For compound **1**, the same EMO was observed using sulfated CDs (S-β-CD, and HDMS-β-CD) and CM-β-CD. These chiral selectors conferred ***1SR***, followed by ***1RS*** migration order. When changing the chiral selector to SBE-β-CD, EMO reversal was obtained with high *R_s_* values. Succ-β-CD also provided the same EMO as SBE-β-CD. Although the obtained enantioresolution value was low with the succinyl derivative, the ***1RS*** > ***1SR*** migration order can clearly be distinguished (Figure 3).

As mentioned above, enantiomers of compound **2** displayed the poorest chiral interactions with the selected CD derivatives, accompanied by low enantioresolution values in most cases, apart from when using HDMS-β-CD as the chiral selector. However, even with the low *R*_s_ values, EMO reversal could be detected, as the *R*-enantiomer migrated first with SBE-β-CD, while the *S*-enantiomer migrated first with CM-β-CD, S-β-CD, and HDMS-β-CD.

Low selectivities were also observed for compound **3** using the CDs selected. Nevertheless, EMO reversal was observed when changing the chiral selector from Succ-β-CD or S-β-CD to SBE-β-CD or CE-β-CD.

In the case of compound **4**, most of the applied β-CD derivatives conferred the ***4S***, followed by ***4R*** migration order, apart from SBE-β-CD and HDMS- β-CD, where the EMO was reversed.

Interestingly, for compounds **5**, and **6**, which were generally characterized with the highest *R*_s_ values among the tested analytes, the EMO remained unchanged, regardless of the β-CD derivative applied.

### 2.4. EMO Reversal Based on the Substitution Pattern of the CDs

Several cases of chiral separations were described, where not only the nature of the substituents but also their location on the CD rim would lead to EMO reversal [36,39,40]. This phenomenon is sometimes also observed when moving from randomly substituted derivatives to single-isomer CDs.

Similar cases were detected for the studied analytes as well. Randomly substituted CDs are characterized by a mixture of multiple isomers with distinct substitution degrees and patterns that contribute to the enantiomer separation process. It has been commonly observed that randomly substituted CDs exhibit greater enantioselectivity compared to the single-isomer derivatives, due to the combined effect of multiple contributing isomers [41,42]. In our case, however, HS-β-CD greatly outperformed S-β-CD in all cases, except for compound **2**. Moreover, out of the six enantiomeric pairs, in two instances, switching from the randomly sulfated, multicomponent mixtures of S-β-CD (sulfation occurs randomly at hydroxyl positions, declared DS: 7–11), to the single isomeric HS-β-CD (in which case sulfation specifically occurs at all hydroxyl groups at position 6), EMO reversal was obtained (compounds **2** and **3**). S-β-CD offered the *S*-enantiomer first migration order for both enantiomeric pairs, while EMO reversal was obtained when changing the chiral selector to HS-β-CD (Figure 4A,B).

Opposite EMOs were also observed in some cases, when comparing the results obtained with the single-isomer CDs, HS-β-CD and HDMS-β-CD. The difference between the two chiral selectors is that, while both are sulfated at position 6, in the latter case the hydroxyls in positions 2 and 3 are methyl-substituted, as opposed to being unsubstituted. In almost all cases, HS-β-CD performed better than HDMS-β-CD, and both CDs provided the same EMO, indicating that generally, the enantioseparation capacity of the CD diminished by the derivatization of the secondary rim. The exemption was compound **2**, where HS-β-CD displayed only very weak enantioselectivity, in the form of a small peak splitting, while HDMS-β-CD offered baseline separation of the enantiomers with *R*_s_ > 2, accompanied by opposite EMO (Figure 4A).

Opposite EMO was also observed for compound **4**, as the application of HS-β-CD resulted in baseline separation of the enantiomers with high *R*_s_ values (*R*_s_ > 5), with the ***4S*** > ***4R*** EMO. Methylation of the chiral selector at positions 2 and 3 lead not only to a drastic decrease in enantioresolution (only a weak peak splitting, *R*_s_ > 0.3), but also a reversal of EMO (Figure 4C) Differences between the EMO observed with HS-β-CD and HDMS-β-CD were also observed by Konjaria and Scriba, when studying the CE enantioseparation of alanyl–phenylalanine analogs with negatively charged CDs [33].

### 2.5. EMO Reversal Based on the Structure of the Studied Oxazolidinones

Four of the six studied analytes share the oxazolidinone core structure. The exceptions are two thio-analogs of compound **5**, specifically compound **4**, which is a benzylthiazolidinethione and compound **6**, which is a benzyloxazolidinethione. Thus, their chemical structure is extremely similar, and is characterized by five-membered heterocycles, having -S-CS-NH- (compound **4**), -O-CO-NH (compound **5**), and -O-CS-NH- units (compound **6**). Considering the marginal dissimilarity between these compounds, it is intriguing to investigate the outcomes obtained from the enantioseparation of these analytes. Before comparing the EMO obtained for these compounds, it is also important to generally look at the obtained enantioresolution values. In general, it can be stated, that replacing both oxygen atoms from the oxazolidinone core structure with sulfur (yielding compound **4**) leads to a decrease in enantioresolution for almost all CDs tested. There were cases where only peak splitting was observed for compound **4**, while compounds **5**, and **6** were baseline resolved (such as CM-β-CD, and S-β-CD). HS-β-CD was the only CD, where compound **4** attained higher enantioresolution values (*R*_s_ = 5.42) than compound **6** (*R*_s_ = 4.63), and almost the same as compound **5** (*R*_s_ = 5.45). However, in this case, as can be observed, high enantioresolution values were obtained for all compounds.

Notably, a similar trend (lower chiral selectivity towards compound **4**) was also observed in our previous publication, using polysaccharide chiral stationary phases in liquid chromatography under polar organic mode to separate compounds **1**–**5** [28]. This dissimilarity in enantiodiscrimination was rationalized by the comparatively greater volume and lesser electronegativity of sulfur, which might influence the three-dimensional spatial structure of the thio-analog and hence its interaction with the chiral selector. Moreover, it is noteworthy that sulfur exhibits a notable tendency towards a more orthogonal orientation towards the donor atom. These variations could culminate in a reduced capacity for enantiorecognition. Although this hypothesis needs further confirmation, it is also notable that from these three enantiomeric pairs, only the thiazolidinethione analog displayed chiral interactions with the larger cavity-sized CM-γ-CD.

When comparing the EMO for these three compounds, it can be observed that for compounds **5** and **6**, the same migration order of the enantiomers was observed for all chiral selectors, with the *S*-enantiomers migrating first, followed by the *R*-enantiomers. However, for compound **4**, with those obtained with compounds **5**, and **6**, two instances of EMO reversals were observed: when SBE-β-CD, was used, ***S*** > ***R*** EMO was observed for compounds **5** and **6**, while opposite EMO was observed for compound **4**. When using HDMS-β-CD as the chiral selector peak splitting was observed for compound **4**, with the ***R*** > ***S*** EMO, while much higher enantioresolution values were obtained for **5** and **6** (see Table 1).

Comparing the results obtained for compounds **5** and **6**, it can be generally stated that the enantiomers of these analytes were most frequently baseline resolved and displayed the highest *R*_s_ values. These compounds differ in the nature of the heteroatom present in position 2: while compound **5** is an oxazolidine-2-on, compound **6** is an oxazolidine-2-thione. From the obtained results, it seems that this change in structure does not affect EMO in these cases.

Compounds **3** and **5** display dissimilarity solely in the presence of a dimethyl moiety at position 3. Interestingly, the dimethyl derivative compound **3** showed chiral interactions with the smaller cavity-sized CM-α-CD, while compound **5** did not. A somewhat opposite case was observed for HDMS-β-CD, where the enantiomers of compound **5** were baseline resolved, while the dimethylated compound **3** showed no chiral interactions with the single isomeric 2,3-*O*-dimethylated sulfated β-CD (HDMS-β-CD). Comparing the separations obtained with the other CDs, where the enantiomers of both compounds showed chiral interactions, several EMO reversals can be identified. As discussed earlier, in the case of compound **5**, the same EMO was observed with all the CDs where at least peak splitting was observed, ***5S*** > ***5R***. In the case of compound **3**, EMO reversal was observed when compared to compound **5**, when using the carboxyalkylated CDs (CM-α-CD, CM-β-CD, CE-β-CD), SBE-β-CD, and HS-β-CD. In these cases, the EMO was ***3R*** > **3*S***. However, baseline separation was obtained only with the latter CD derivate. In general compound **5** provided the highest enantioresolution values, while compound **2** was characterized in general with some of the lowest *R*_s_ values; thus, it can be stated that dimethylation of the oxazolidinone core greatly reduces the possibilities of chiral interactions with the selected CDs.

Another comparison, which is worthy of taking another look at, is that of compounds **2** and **5**. Both molecules share the same oxazolidinone core structure, substituted at position 4; however, compound **2** is phenyl-substituted, while compound **5** is benzyl-substituted. Although the difference between the two compounds is a single methylene group, differences were observed both in enantioresolution values and EMO in some cases. Generally, higher *R*_s_ values were obtained for compound **5**. Interestingly, enantiomers of compound **5** were baseline separated with Succ-β-CD, while no enantiorecognition was observed for compound **2**. However, when using CM-γ-CD, enantiodiscrimination was only observed for compound **2**. Several cases of EMO reversals were also identified for these compounds when using SBE-β-CD and HS-β-CD. In both cases, much higher enantioresolution values were obtained for compound **5**, than compound **2**.

## 3. Materials and Methods

### 3.1. Materials

Enantiopure (4*R*,5*S*)-4-Methyl-5-phenyl-2-oxazolidinone (***1RS***), (4*S*,5*R*)-4-Methyl-5-phenyl-2-oxazolidinone (***1SR***), (*R*)-4-Phenyl-2-oxazolidinone (***2R***), (*S*)-4-Phenyl-2-oxazolidinone (***2S***), (*R*)-4-Benzyl-5,5-dimethyl-2-oxazolidinone (***3R***), (*S*)-4-Benzyl-5,5-dimethyl-2-oxazolidinone (***3S***), (*R*)-4-benzylthiazolidine-2-thione (***4R***), (*S*)-4-benzylthiazolidine-2-thione (***4S***), (*R*)-4-Benzyl-2-oxazolidinone (***5R***), (*S*)-4-benzyl-2-oxazolidinone (***5S***), (*R*)-4-benzyloxazolidine-2-thione (***6R***), and (*S*)-4-benzyloxazolidine-2-thione (***6S***) were purchased from Sigma-Aldrich Hungary (Budapest, Hungary).

Methanol (MeOH) was purchased from Merck (Darmstadt, Germany). NaOH, HCl, phosphoric acid, and phosphate salts (Sigma-Aldrich, Budapest, Hungary) used for the preparation of preconditioning and buffer solutions were of analytical grade. All reagents were used without further purification. Ultrapure water was used throughout this study, prepared with a Barnstead Nanopure Diamond water purification system (Boston, MA, USA).

The following CD derivatives with various DS were obtained from Cyclolab (Budapest, Hungary): CM-α-CD DS~3.5, CM-β-CD DS~3.5, CM-γ-CD DS~3.5, CE-β-CD DS~3.5, Succ-β-CD DS~3.5, SBE-β-CD DS~6.3; as well as the following single isomeric CDs: heptakis-(6-O-sulfo)-β-CD (HS-β-CD) heptakis-(2,3-di-O-methyl-6-O-sulfo)-β-CD (HDMS-β-CD). S-β-CD with a declared range of DS of 7–11 was from Sigma-Aldrich Hungary (Budapest, Hungary).

### 3.2. Preparation of BGE and Sample Solutions

Experiments were performed in phosphate buffer pH 6.0. In detail, 50 mM KH_2_PO_4_ solution was prepared; the BGEs contained CDs in various concentrations (at least 5 concentrations, over the range of 2.5–25 mM; where the generated currents permitted it, 50 mM CD concentration was also employed. In this case, the separation voltage was lowered to 15 kV). After the addition of the appropriate amount of CD, the pH of the BGEs was adjusted to pH = 6.00 ± 0.05 with NaOH 0.1 M.

Stock solutions of around 2 mg/mL of each analyte were prepared in MeOH. Sample solutions were prepared by further diluting the stock solutions with water. During the analyses, sample solutions with an enantiomeric ratio of 1:3 were used, to track the EMO of the separation.

All BGE and sample solutions were filtered through 0.45 µm regenerated cellulose (RC) syringe filters from BGB Analytik (Schloßböckelheim, Germany).

### 3.3. Enantioselective CE Conditions

CE experiments were performed on an Agilent G1600 CE instrument (Agilent Technologies, Waldbronn, Germany), equipped with a photodiode array detector and ChemStation software for data handling. An untreated fused-silica capillary (50 µm I.D., 363 µm O.D., 48.5 cm total, and 40 cm effective length) from Agilent (Waldbronn, Germany) was used throughout the study. New capillaries were conditioned by flushing with 1 M NaOH for 20 min followed by 0.1 M NaOH for 20 min and water for 10 min. The same preconditioning procedure was applied at the start of each working day. Prior to all runs the capillary was preconditioned by flushing with 0.1 M NaOH (2 min), water (1 min), and BGE (2 min). UV detection was performed at 220 nm. Injections were carried out by applying a pressure of 50 mbar for 3 s. The separation voltage was 20 kV, except for cases where 50 mM CD concentrations were tested, in which case the separation voltage was lowered to 15 kV. The generated currents were kept below 100–110 μA, and the capillary temperature was maintained at 25 °C.

## 4. Conclusions

In this study, the enantioseparation of four neutral oxazolidinones and two related thio-analogs was performed using nine different anionic CDs as chiral selectors at varying concentrations. A special focus was on observing the reversal EMO, which was found to occur based on the cavity size, substituent nature, and substitution pattern of CDs. The versatility of CDs for fine-tuning enantioseparations was highlighted by the significant differences in enantiomer resolution values observed across the different separation systems and the EMO reversals observed. Additionally, in several cases, even minute changes in the structure of the analytes lead to interesting cases of EMO reversals, which further underlines the importance of specific selector–selectand interactions.

Out of the nine anionic CDs, the single-isomer sulfated CD, HS-β-CD provided in general the highest enantioresolution values for the studied enantiomeric pairs, followed by the also single-isomer HDMS-β-CD. The least successful enantioseparations were observed with CM-α-CD and CM-γ-CD, indicating that the β-CD cavity is probably the most suitable to accommodate the studied analytes, and both smaller and larger cavities result in a decrease in enantiodiscrimination.

Given the importance of the chirality of oxazolidinones widely applied as both chiral auxiliaries and pharmacologically active core structures, enantioseparation of this class of compounds is extremely important for their enantiomeric quality control. Herein, the first comprehensive chiral CE study was performed for this class of compounds.

## Figures and Tables

**Figure 1 molecules-28-04530-f001:**
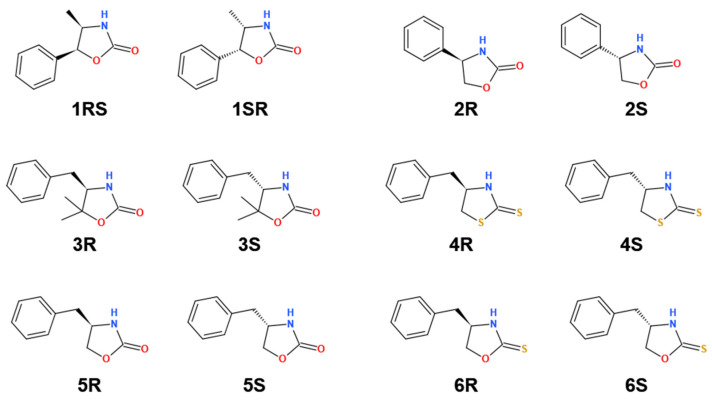
2D structures of the studied analytes.

**Figure 2 molecules-28-04530-f002:**
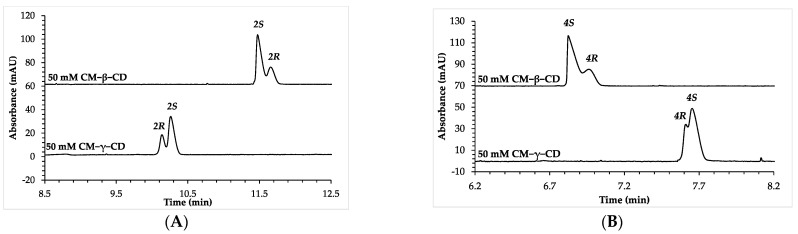
Electropherograms depicting the cavity-size-dependent EMO reversal for (**A**) compound **1**, and (**B**) compound **2**. The used chiral selectors and their concentrations are shown on the individual traces. Other CE conditions: 48.5/40 cm, 50 μm I.D. (363 μm O.D.) uncoated, fused-silica capillary; 50 mM phosphate buffer, supplemented with a chiral selector as shown on the individual traces, pH 6.0. Capillary temperature: 25 °C, separation voltage: 15 kV (generated currents: around 95 μA in all cases); hydrodynamic injection: 50 mbar × 3 s, detection at 210 nm.

**Figure 3 molecules-28-04530-f003:**
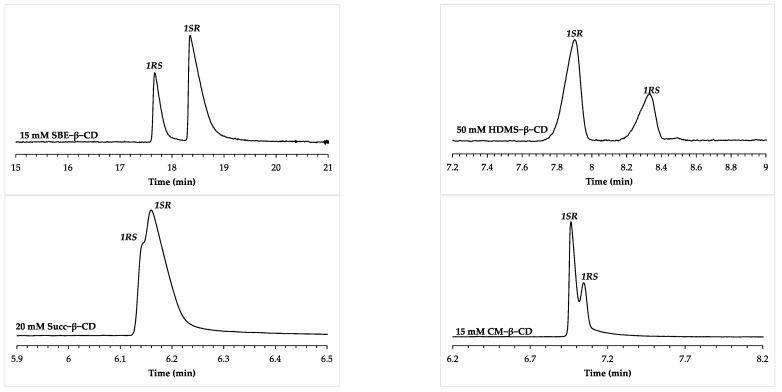
Electropherograms depicting the examples of substituent-dependent EMO reversal for compound **1**. The used chiral selectors and their concentrations are shown on the individual traces. Separation voltage: 20 kV (15 kV for HDMS-β-CD). Generated currents were between ~50 μA for Succ-β-CD, and ~100 μA for HDMS-β-CD. Other CE conditions are the same as in Figure 2.

**Figure 4 molecules-28-04530-f004:**
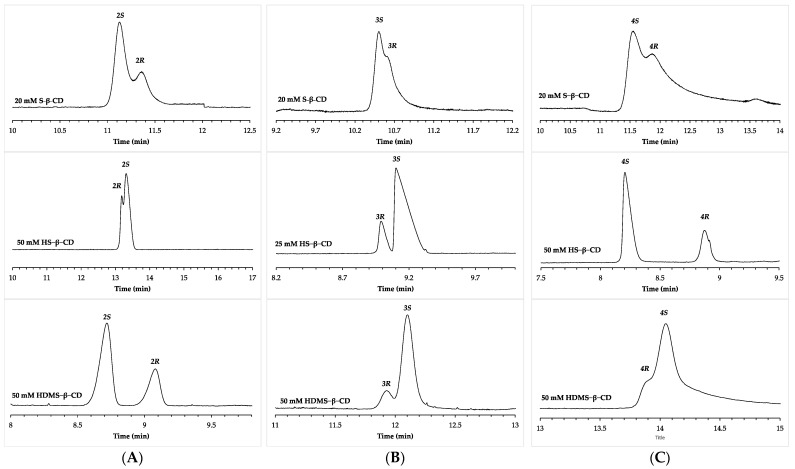
Comparative electropherograms obtained with randomly substituted S-β-CD and single-isomer HS-β-CD and HDMS-β-CD for the enantiomers of compounds (**A**) **2**, (**B**) **3**, and (**C**) **4**. Separation voltage: 20 kV (15 kV for 50 mM HS-β-CD, and 50 mM HDMS-β-CD). Generated currents were between ~75 and ~100 μA. Other CE conditions are the same as in Figure 2.

**Table 1 molecules-28-04530-t001:** Enantiomeric resolution values and the observed EMO of the studied compounds by employing the indicative chiral selector.

CD\Analyte		1	2	3	4	5	6
CM-α-CD	EMO	-	-	R > S	-	-	-
	*R*_s_ max.	-	-	0.53	-	-	-
CM-β-CD	EMO	SR > RS	S > R	R > S	S > R	S > R	S > R
	*R*_s_ max.	1.04	1.11	0.89	0.94	2.82	2.00
CM-γ-CD	EMO	-	R > S	-	R > S	-	-
	*R*_s_ max.	-	0.88	-	0.58	-	-
CE-β-CD	EMO	-	-	R > S	S > R	S > R	S > R
	*R*_s_ max.	-	-	0.30	0.52	0.74	0.81
SBE-β-CD	EMO	RS > SR	R > S	R > S	R > S	S > R	S > R
	*R*_s_ max.	2.97	<0.50	<0.50	0.62	1.31	3.23
Succ-β-CD	EMO	RS > SR	-	S > R	S > R	S > R	S > R
	*R*_s_ max.	0.74	-	1.17	1.72	1.80	2.19
S-β-CD	EMO	SR > RS	S > R	S > R	S > R	S > R	S > R
	*R*_s_ max.	2.44	0.87	<0.50	<0.50	2.02	1.55
HS-β-CD	EMO	SR > RS	R > S	R > S	S > R	S > R	S > R
	*R*_s_ max.	3.04	0.53	1.60	5.42	5.45	4.63
HDMS-β-CD	EMO	SR > RS	S > R	R > S	R > S	S > R	S > R
	*R*_s_ max.	2.50	2.23	0.91	<0.50	1.58	1.26

## Data Availability

All relevant data are included in the article.

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
