# Peer review of "Chiral Separation of Oxazolidinone Analogs by Capillary Electrophoresis Using Anionic Cyclodextrins as Chiral Selectors: Emphasis on Enantiomer Migration Order"

_molecules, 2023, doi:10.3390/molecules28114530_

Round 1

Reviewer 1 Report

In this manuscript, using cyclodextrin (CD) derivatives as chiral selectors, comparative chiral separations of enantiomeric pair of four oxazolidinone and two related thio-derivatives were performed by capillary electrophoresis. However, the provided method was also complex preparation, separation and analysis procedure. The content of the manuscript is lacking innovation and not well structured.

Below is presented the points that the authors should take into account in the manuscript:

1. The introduction was too long and lacks focus.

2. Method validation should be described in detail in Results section, such as reproducibility and stability.

3. what the meaning of the α-、β-、γ-CD selectors design? β-CD has been recognized as the best chiral selective agent.

4. A large number of experiments have been done for EMO reversal, but not explaining the Phenomenon and mechanism.

5. The method should be well demonstrated by analyzing real samples. The present situation is not satisfactory at this point.

Author needs to carefully check grammatical throughout the manuscript. Some sentences were hardly to understand.

Author needs to carefully check grammatical throughout the manuscript. Some sentences were hardly to understand.

Author Response

In this manuscript, using cyclodextrin (CD) derivatives as chiral selectors, comparative chiral separations of enantiomeric pair of four oxazolidinone and two related thio-derivatives were performed by capillary electrophoresis. However, the provided method was also complex preparation, separation and analysis procedure. The content of the manuscript is lacking innovation and not well structured.

 Below is presented the points that the authors should take into account in the manuscript:

 1. The introduction was too long and lacks focus.

Several, shorter parts of the introduction section were deleted. The introduction was restructured for further clarity. As suggested by the Editor, a separate paragraph was also introduced, describing some of the chiral CE methods already available for oxazolidinone analogs.

  1. Method validation should be described in detail in Results section, such as reproducibility and stability

The main purpose of the present study was the comparative enantioseparation of the oxazolidinones and related thio-analogs, with a special emphasis on enantiomer migration order reversals. These types of manuscripts offer valuable information for further in-depth exploration of the enantiodiscrimination mechanisms responsible for the observed EMO reversals and also offer a good starting point for chiral method development for oxazolidinone analogs.

Since many separation systems were tested, for a high number of analytes, method optimization, and method validation were never the scope of this study, but rather a general overview of these separation systems.

  1. what the meaning of the α-、β-、γ-CD selectors design? β-CD has been recognized as the best chiral selective agent.

 While β-CD derivatives often provide superior enantiodiscrimination, compared to the corresponding α-, and γ-CD counterparts, this is not always the case. There are a great number of studies where the latter CD derivatives were the better choice.

Moreover, as observed in our case, switching only the cavity size of the CD derivative could also lead to EMO reversal, which is sometimes needed to achieve the favorable, distomer-first migration order.

In our case, based on the results obtained with the different cavity-sized carboxymethylated-CDs, indeed it seemed that β-CD derivatives are the most promising chiral selectors, thus, further CD derivatives were selected from this family.

  1. A large number of experiments have been done for EMO reversal, but not explaining the Phenomenon and mechanism.

Based on the available literature data, EMO reversals were most probably almost exclusively the results of structural changes in the transient diastereomer complexes. Tracking the structural changes responsible for some of the EMO reversals through NMR spectroscopy and molecular modeling are in the plan for a future manuscript. 

  1. The method should be well demonstrated by analyzing real samples. The present situation is not satisfactory at this point.

There are no classical real samples for the analytes in this study, the used substances are in fact the real samples per se. These are not used as active pharmaceutical ingredients but in asymmetric syntheses.

6. Author needs to carefully check grammatical throughout the manuscript. Some sentences were hardly to understand.

English language correction was undertaken to enhance the overall readability of the manuscript.

Reviewer 2 Report

This study aimed at using nine different anionic CDs as chiral selectors at varying concentrations for the enantioseparation of four neutral oxazolidinones and two related thio-analogs. It was found that HS-β-CDs had the best enantioseparation performance. The experiments were well designed and the results were clearly presented. Moreover, the reversal EMO by cavity size of CDs, substituent nature of CDs, substitution pattern of CDs and even minute changes in the structure of the analytes were also discussed in detail. Therefore, I think this work can be accepted for publication after minor revision of the following issues:

1.     The enantioseparation performance of anionic CDs and cationic CDs can be simply investigated.

2.     The mechanism of the influence of chiral separation and EMO reversal should be further discussed, for example by molecular simulation.

3.     Whether the enantioseparation of each cyclodextrin in this study was compared under their respective optimal separation conditions.

4.     Is this study research the separation stability and reproducibility of HS-β-CDs?

5.     In view of the similar structure of oxazolidinones, can HS-β-CDs realize the efficient separation of multiple enantiomers?

The manuscript contains some grammar or syntax errors. An improvement of the English language is necessary. For example, "The group of synthetic antibiotics were" should be "The group of synthetic antibiotics was" to match the singular noun "group."

Author Response

This study aimed at using nine different anionic CDs as chiral selectors at varying concentrations for the enantioseparation of four neutral oxazolidinones and two related thio-analogs. It was found that HS-β-CDs had the best enantioseparation performance. The experiments were well designed and the results were clearly presented. Moreover, the reversal EMO by cavity size of CDs, substituent nature of CDs, substitution pattern of CDs and even minute changes in the structure of the analytes were also discussed in detail. Therefore, I think this work can be accepted for publication after minor revision of the following issues:

 1. The enantioseparation performance of anionic CDs and cationic CDs can be simply investigated.

 Thank you for your comment. In the case of neutral compounds like oxazolidinones, charged cyclodextrin derivatives are necessary for their investigation. Considering their accessibility and wide range of applications, we have chosen anionic cyclodextrins for the current investigation. It is important to note that cyclodextrins with positive charges can interact with the capillary wall, which is a significant factor to consider. However, in future studies, cationic CDs can also be utilized for further exploration.

  1. The mechanism of the influence of chiral separation and EMO reversal should be further discussed, for example by molecular simulation.

Based on the available literature data, EMO reversals were most probably almost exclusively the results of structural changes in the transient diastereomer complexes. Tracking the structural changes responsible for some of the EMO reversals through NMR spectroscopy and molecular modeling are in the plan for a future manuscript. 

  1. Whether the enantioseparation of each cyclodextrinin this study was compared under their respective optimal separation conditions.

The same separation conditions were applied for all CD derivatives. The enantioresolution results presented in Table 1 represent the maximal resolution values obtained across the concentration range of the chiral selector. In most instances, the highest Rs values were obtained at the highest CD concentrations, except for the carboxyalkylated CDs, where enantioresolution values attained a maximum with increasing CD concentration, and after that maximum, Rs values decreased with increasing concentrations. 

  1. Is this study research the separation stability and reproducibility of HS-β-CDs?

 All sulfated CDs, including HS-β-CD were kept in the freezer when not in use. During the time span of this study (approx. 6 months), no issues were observed regarding the stability of the chiral selector. Separations could be and were reproduced successfully throughout the study, with no issues at all.

  1. In view of the similar structure of oxazolidinones, can HS-β-CDs realize the efficient separation of multiple enantiomers?

 Unfortunately, under the conditions tested, HS-β-CD was unable to baseline separate the enantiomers of compound 2. When reviewing the electropherograms, it was clear that using the present conditions, peak overlapping would occur between for ex. compounds 1SR, 4S, and 5S; compounds 1RS, and 3R; and compounds 3S and 4R. Further optimization could probably lead to simultaneous baseline separation of all compounds, however, this was outside the scope of the present manuscript.

6. The manuscript contains some grammar or syntax errors. An improvement of the English language is necessary. For example, "The group of synthetic antibiotics were" should be "The group of synthetic antibiotics was" to match the singular noun "group."

English language correction was undertaken for the entire manuscript to enhance the overall readability of the manuscript.

Reviewer 3 Report

This manuscript (MS) deals with enantioseparations of six oxazolidinone analogs by capillary electrophoresis (CE) using a series of nine derivatized negatively charged alpha-, beta- and gamma-cyclodextrins (CDs) as chiral selectors. The separation mechanism and the enantiomer migration order (EMO) were studied with respect to the structure and physicochemical properties of both the analytes and CDs. The MS brings new interesting results. It can be accepted for publication after the following revision.

1.      Line 31: “EMO was observed” > “EMO reversal was observed” or “reverse EMO was observed”.

2.      Line 79: The abbreviation FDA should be explained when first used.

3.      Line 100: A general term “efficiency” should be changed to the particular term “separation efficiency”.

4.      Line 100: “fast analysis time” > “fast analysis” or “short analysis time”

5.      Line 107: For the statement that CDs are the preferred chiral selectors used in CE, the following papers should be cited:

P. Rezanka, K. Navratilova, M. Rezanka, V. Kral, and D. Sykora. Application of cyclodextrins in chiral capillary electrophoresis. Electrophoresis 35 (19):2701-2721, 2014.

G. K. E. Scriba. Chiral recognition in separation sciences. Part I: Polysaccharide and cyclodextrin selectors. Trends Anal.Chem. 120:115639, 2019.

I. Fejos, E. Kalydi, M. Malanga, G. Benkovics, and S. Beni. Single isomer cyclodextrins as chiral selectors in capillary electrophoresis. J.Chromatogr.A 1627:461375, 2020.

P. Peluso and B. Chankvetadze. Native and substituted cyclodextrins as chiral selectors for capillary electrophoresis enantioseparations: Structures, features, application, and molecular modeling. Electrophoresis 42 (17-18):1676-1708, 2021.

6.      Line 151 and elsewhere: A complete composition of the phosphate buffer (including the cationic component and its concentration) should be presented.

7.      Line 152 and elsewhere: General term “applied voltage” should be changed to the specific term “separation voltage” that is correctly used in line 154.

8.      Line 207: In addition to capillary I.D., capillary O.D. should be presented.

9.      Line 209: In addition to separation voltage, the values of electric current in the presented CE experiments should be given.

10.  Line 209 and elsewhere: In the SI unit system, the correct symbol for the time unit second is “s” not “sec”.

11.  Line 212: In addition to refs. [27-33], the following papers should be cited:

D. Koval, L. Severa, L. Adriaenssens, J. Vavra, F. Teply, and V. Kasicka. Chiral analysis of helquats by capillary electrophoresis: Resolution of helical N-heteroaromatic dications using randomly sulfated cyclodextrins. Electrophoresis 32 (19):2683-2692, 2011.

M. Ruzicka, D. Koval, J. Vavra, P. E. Reyes-Gutierrez, F. Teply, and V. Kasicka. Interactions of of helquats with chiral acidic aromatic analytes investigated by partial-filling affinity capillary electrophoresis. J.Chromatogr.A 1467:417-426, 2016.

12.  In Figs. 2-4, the labeling of axes, peaks and CE records should be enlarged. Current labeling is not well readable.

13.  In Fig. 4C, top record, the enantiomers are wrongly indicated as 2S and 2R instead of 4S and 4R.

14.  Line 404: id > I.D. (in accordance with line 207).

Author Response

This manuscript (MS) deals with enantioseparations of six oxazolidinone analogs by capillary electrophoresis (CE) using a series of nine derivatized negatively charged alpha-, beta- and gamma-cyclodextrins (CDs) as chiral selectors. The separation mechanism and the enantiomer migration order (EMO) were studied with respect to the structure and physicochemical properties of both the analytes and CDs. The MS brings new interesting results. It can be accepted for publication after the following revision.

  1. Line 31: “EMO was observed” > “EMO reversal was observed” or “reverse EMO was observed”.

Corrected.

  1. Line 79: The abbreviation FDA should be explained when first used.

 Corrected.

  1. Line 100: A general term “efficiency” should be changed to the particular term “separation efficiency”.

 Corrected.

  1. Line 100: “fast analysis time” > “fast analysis” or “short analysis time”

 Corrected.

 5. Line 107: For the statement that CDs are the preferred chiral selectors used in CE, the following papers should be cited:

 Rezanka, K. Navratilova, M. Rezanka, V. Kral, and D. Sykora. Application of cyclodextrins in chiral capillary electrophoresis. Electrophoresis 35 (19):2701-2721, K. E. Scriba. Chiral recognition in separation sciences. Part I: Polysaccharide and cyclodextrin selectors. Trends Anal.Chem. 120:115639, Fejos, E. Kalydi, M. Malanga, G. Benkovics, and S. Beni. Single isomer cyclodextrins as chiral selectors in capillary electrophoresis. J.Chromatogr.A 1627:461375, Peluso and B. Chankvetadze. Native and substituted cyclodextrins as chiral selectors for capillary electrophoresis enantioseparations: Structures, features, application, and molecular modeling. Electrophoresis 42 (17-18):1676-1708, 2021.

The suggested references were introduced in the revised manuscript.

 6. Line 151 and elsewhere: A complete composition of the phosphate buffer (including the cationic component and its concentration) should be presented.

 This information is now included in the revised manuscript.

  1. Line 152 and elsewhere: General term “applied voltage” should be changed to the specific term “separation voltage” that is correctly used in line 154.

 Corrected throughout the manuscript.

  1. Line 207: In addition to capillary I.D., capillary O.D. should be presented.

 Corrected throughout the manuscript.

  1. Line 209: In addition to separation voltage, the values of electric current in the presented CE experiments should be given.

 Corrected.

  1. Line 209 and elsewhere: In the SI unit system, the correct symbol for the time unit second is “s” not “sec”.

 Corrected.

  1. Line 212: In addition to refs. [27-33], the following papers should be cited: Koval, L. Severa, L. Adriaenssens, J. Vavra, F. Teply, and V. Kasicka. Chiral analysis of helquats by capillary electrophoresis: Resolution of helical N-heteroaromatic dications using randomly sulfated cyclodextrins. Electrophoresis 32 (19):2683-2692, 2011, Ruzicka, D. Koval, J. Vavra, P. E. Reyes-Gutierrez, F. Teply, and V. Kasicka. Interactions of of helquats with chiral acidic aromatic analytes investigated by partial-filling affinity capillary electrophoresis. J.Chromatogr.A 1467:417-426, 2016.

 The first reference, which is relevant to the topic, was introduced in the revised version of the manuscript.

  1. In Figs. 2-4, the labeling of axes, peaks and CE records should be enlarged. Current labeling is not well readable.

 Corrected.

  1. In Fig. 4C, top record, the enantiomers are wrongly indicated as 2S and 2R instead of 4S and 4R.

 Corrected.

  1. Line 404: id > I.D. (in accordance with line 207).

Corrected.

Round 2

Reviewer 1 Report

Improve the English expression.

Improve the English expression.